# Advancements in Oncology with Artificial Intelligence—A Review Article

**DOI:** 10.3390/cancers14051349

**Published:** 2022-03-06

**Authors:** Nikitha Vobugari, Vikranth Raja, Udhav Sethi, Kejal Gandhi, Kishore Raja, Salim R. Surani

**Affiliations:** 1Department of Internal Medicine, Medstar Washington Hospital Center, Washington, DC 20010, USA; nikitha.vobugari@medstar.net (N.V.); kejal.d.gandhi@medstar.net (K.G.); 2Department of Medicine, P.S.G Institute of Medical Sciences and Research, Coimbatore 641004, Tamil Nadu, India; drvikranthraja@gmail.com; 3School of Computer Science, University of Waterloo, Waterloo, ON N2L 3G1, Canada; udhav.sethi@uwaterloo.ca; 4Department of Pediatric Cardiology, University of Minnesota, Minneapolis, MN 55454, USA; drkishoreraja@gmail.com; 5Department of Pulmonary and Critical Care, Texas A&M University, College Station, TX 77843, USA

**Keywords:** artificial intelligence, machine learning, deep learning, convolutional neural networks, support vector machine, breast oncology, brain tumors, colon cancer

## Abstract

**Simple Summary:**

With the advancement of artificial intelligence, including machine learning, the field of oncology has seen promising results in cancer detection and classification, epigenetics, drug discovery, and prognostication. In this review, we describe what artificial intelligence is and its function, as well as comprehensively summarize its evolution and role in breast, colorectal, and central nervous system cancers. Understanding the origin and current accomplishments might be essential to improve the quality, accuracy, generalizability, cost-effectiveness, and reliability of artificial intelligence models that can be used in worldwide clinical practice. Students and researchers in the medical field will benefit from a deeper understanding of how to use integrative AI in oncology for innovation and research.

**Abstract:**

Well-trained machine learning (ML) and artificial intelligence (AI) systems can provide clinicians with therapeutic assistance, potentially increasing efficiency and improving efficacy. ML has demonstrated high accuracy in oncology-related diagnostic imaging, including screening mammography interpretation, colon polyp detection, glioma classification, and grading. By utilizing ML techniques, the manual steps of detecting and segmenting lesions are greatly reduced. ML-based tumor imaging analysis is independent of the experience level of evaluating physicians, and the results are expected to be more standardized and accurate. One of the biggest challenges is its generalizability worldwide. The current detection and screening methods for colon polyps and breast cancer have a vast amount of data, so they are ideal areas for studying the global standardization of artificial intelligence. Central nervous system cancers are rare and have poor prognoses based on current management standards. ML offers the prospect of unraveling undiscovered features from routinely acquired neuroimaging for improving treatment planning, prognostication, monitoring, and response assessment of CNS tumors such as gliomas. By studying AI in such rare cancer types, standard management methods may be improved by augmenting personalized/precision medicine. This review aims to provide clinicians and medical researchers with a basic understanding of how ML works and its role in oncology, especially in breast cancer, colorectal cancer, and primary and metastatic brain cancer. Understanding AI basics, current achievements, and future challenges are crucial in advancing the use of AI in oncology.

## 1. Introduction

Artificial intelligence (AI) is a field in which computers are programmed to mimic human intelligence. The abundance of data in the field of medicine makes it a good candidate for problem solving using machine learning (ML). In oncology, ML can be used to diagnose and classify tumors, detect early-stage tumors, gather genetic and histopathological data, assist in pre- and post-operative planning, and predict overall survival outcomes [1]. Deep Learning (DL), a type of ML, has proven to be effective in automating time-consuming steps such as detection and segmentation of lesions [2,3,4].

AI-based models have demonstrated excellent accuracy rates of cancer detection on screening mammography and breast cancer (BC) prediction based on genetics and hormonal factors [5,6,7]. AI plays a crucial role in early detection, classification, histopathological aspects, genetics, and molecular markers detection in colorectal cancer (CRC) [8,9,10]. As a result of extensive data in present-day screening and improvements in life expectancy caused by early detection of breast and colon cancer, we review the potential of AI-based diagnostics and therapeutics. Because mammograms and colonoscopies are widely used in the general population worldwide, AI can be used extensively in future studies on cancer screening to build generalizable AI systems [11]. AI has made its way into other cancer types, which we do not review here. For instance, lung cancer screening is reserved for smokers, and the United States Preventive Services Task Force (USPSTF) approved low-dose chest computed tomography (CT) scans in 2013, and prostate cancer screening has not yet been approved universally [11,12]. CNS cancers are relatively rare and have a poor prognosis. Studying AI in such rare tumors can provide a scope of precision of AI integration in improving the current standard management. In the area of central nervous system (CNS) tumors, AI and radiomics have notably enhanced detection rates and reduced several time-consuming steps in glioma grading, pre- and intraoperative planning, and postoperative follow-up [13,14,15].

This review article outlines how AI works in simple terminology that medical professionals can understand, how it has improved breast cancer screening, colon polyp detection, and colorectal cancer screening, as well as the implications it has in the management of CNS tumors. A literature search was conducted on PubMed, Google Scholar, arXiv, and Scopus. This is not a systematic review but a narrative review of the literature. We conclude with existing obstacles and future speculations of standardizing AI screening in oncology, as well as proposals for integrating AI basics into medical school curricula.

## 2. How Does Artificial Intelligence Work?

AI is a broad concept that aims to simulate human cognitive ability. ML, an approach to AI, is the study of how computer systems can learn to perform a task or predict an outcome without being explicitly programmed [16]. Mitchell et al. (1997) succinctly defines this learning process as follows: A computer program is said to learn from experience (E) with respect to some class of tasks (T) and performance measure (P), if its performance at tasks in T, as measured by P, improves with experience E. A simple example of such a task is the classification of suspicious abnormality on a screening mammogram as probable malignant or benign [17]. To learn to perform this task, a computer program would experience a dataset containing examples of correctly classified cases of benign and malignant breast lesions and come up with a model that can generalize beyond these data. Its ability to then classify previously unseen examples of breast lesions correctly would be evaluated through a quantitative measure of its performance, such as accuracy, sensitivity, and specificity.

### 2.1. Subtypes of Machine Learning

Algorithms for ML are typically categorized into supervised, unsupervised, or reinforcement learning. Supervised learning algorithms experience a dataset that contains a label (or correct answer) for each data point. Examples of supervised learning algorithms include support vector machine (SVM) [18,19], linear regression, logistic regression, and k-nearest neighbors [20,21]. In contrast, unsupervised algorithms such as k-means clustering [22,23], affinity propagation [24], and gaussian mixture model [25] study a dataset that does not contain labels and learn to derive structure from the given data. A reinforcement learning system trains an agent to behave in an environment by assigning it with a reward for desired behaviors or penalizing it for undesired ones. The overall objective of an ML algorithm can be interpreted as learning an approximate function of the data. This function should take as input a set of features that describe the data and output a prediction corresponding to the learning task. Classical ML algorithms are generally good at approximating linear or simple non-linear functions [13,26].

### 2.2. Deep Learning

DL is a type of ML that enables the learning of complex non-linear functions of the data. Most modern DL methods use neural networks as their learning model, which are loosely inspired by neuroscience [27]. The fundamental computational unit of a neural network is called a neuron. It computes a weighted sum of its inputs and then applies a non-linear operation (often called the activation function) to the sum to compute the output (See Figure 1a). Common activation functions include sigmoid, tanh, and rectified linear activation unit (ReLU) functions. A neural network comprises one or more layers of neurons, with each layer feeding on the outputs of the previous layer. Information flows forward through the network from the input, through a series of intermediate layers (called hidden layers) and finally to the output (see Figure 1b). As the number of layers and units within a layer increase, a neural network can represent functions of increasing complexity. This architecture gives neural networks the ability to learn their own complex features instead of being constrained to the hand-picked features provided as input to the model.

During training, the parameters of the neural network are learned in order to fit the dataset for a given task. This corresponds to minimizing some notion of a cost function, which measures the model’s performance on the task. After each forward pass through the network, the cost function is used to compute the error between the predicted and expected output. An algorithm called backpropagation allows this cost information to flow backward through the neural network while adjusting the network parameters. Backpropagation computes the gradients of the cost function with respect to the network parameters, which determine the level of adjustment to be made to the parameters in each iteration [28]. These gradients are then used to update the network parameters using an optimization algorithm such as stochastic gradient descent (SGD) [29,30].

Apart from the simple feed-forward model discussed above, there are other specialized architectures of neural networks suited for specific tasks. For instance, convolutional neural networks (CNNs) have a grid-like topology and are well suited to process two or three-dimensional inputs such as images [31]. CNNs are designed to capture spatial context and learn correlations between local features, due to which they yield superior performance on image tasks, such as the classification of breast lesions in a screening mammogram as probable malignant or benign (See Figure 1c). CNN-based architectures have also been applied to biomedical segmentation applications [32]. However, CNNs face computational and memory efficiency limitations in three-dimensional (3D) segmentation tasks. More efficient methods have been proposed for the segmentation of 3D data, such as magnetic resonance imaging (MRI) volumes [33]. A recent architecture, occupancy networks for semantic segmentation (OSS-Net) [34], is built upon occupancy networks (O-Net) and contains efficient representations for 3D geometry, which allows for more accurate and faster 3D segmentation [35].

Another family of neural networks, called recurrent neural networks (RNNs), are designed to operate on sequential data. RNNs are well equipped to process sequential inputs of variable lengths for tasks such as machine translation and language modeling. Long Short Term Memory networks (LSTMs) are a special kind of RNNs capable of learning long-term dependencies between inputs [36]. Another technique called attention allows a model to selectively focus on parts of the input data as needed by enhancing specific parts of the input and diminishing others [37]. Recently, a network architecture called the Transformer has achieved state-of-the-art performance in a number of machine learning tasks [38]. Transformers discard recurrence and convolutions entirely, instead relying exclusively on attention mechanisms. Attention-based transformers have demonstrated state-of-the-art segmentation performance and may prove relevance to the field of oncology [39].

## 3. Breast Cancer

BC is the most prevalent cancer originally reported in National Cancer Institute Statistics, 2020 [40]. BC is a major cause of cancer-related mortality after lung cancer [41]. The death rates of BC have decreased annually from 1989 to 2017, attributed to the advancements in screening and therapies [41]. AI has shown enormous benefits in screening mammograms, BC predictive tools formulation, and drug development [5,6,42,43,44].

### 3.1. Screening Mammogram

A screening mammogram is one of the most widely performed screening tests, but these mammograms have limitations of very high false positive and false negative rates [14,42]. The AI models reduced the workload and resulted in a 69% reduction in false positive rates and a higher sensitivity rate in screening mammograms [2,42]. AI in BC screening has good accuracy rates with some methodological issues and evidence gaps [14,45]. 

In the context of mammography, DL algorithms such as CNNs are principally used; the mechanism of the algorithm is illustrated in Figure 1c. The performance of AI is measured by sensitivity, specificity, the area under the curve (AUC), and computation time [46]. Different DL models have been studied with various classification systems to identify abnormalities in mammograms, with overall sensitivity rates ranging from 88% to 96% [47,48,49]. Detection rates are augmented by the positive reinforcement of an AUC over 0.96 after biopsy confirmation [50]. A new AI model from Transpara 1.4.0 screenpoint medical BV, Nijmegen, the Netherlands, expedites interpretation and reduces workload by 20–50% by excluding mammograms with a low likelihood of cancer, allowing radiologists to concentrate on challenging cases [2,51]. The detection performance of radiologists using AI-aided systems was compared to radiologists using conventional systems. Radiologists with AI-aided systems achieved higher AUC rates, sensitivity, and classification performance [52,53]. 

Conventional computer-aided detection (CADe) in mammograms is hampered by high false positive and false negative rates. AI-based CAD systems have proven to reduce false positive rates by 69% and increase in sensitivity ranging from 84% to 91% [42,54]. The concept of double readers (mammogram read by two radiologists independently or together) is used in Europe to reduce false positives and false negatives. The use of AI in place of the second reader maintained a non-inferior performance and reduced the workload by 88% in a simulation study [55]. In another study, a single radiologist assessment was combined with an AI algorithm achieved higher interpretative accuracy with a specificity of 92% vs. 85.9% of a single radiologist’s interpretation. However, any single AI algorithm did not outperform radiologists’ accuracy rates [14]. Double readers are not a standard practice in the United States, but a prospect of cost-effective AI integration with radiologists can increase overall sensitivity. However, the acceptable miss rate threshold should be carefully considered. Another study used the breast imaging reporting and data system (BI-RADS) to incorporate radiologists’ subjective thresholds while using evidence-based data to train AI. The study showed a reduction in false positives by 47.3% and a slight increase in false negatives by 26.7% [56]. AI also has the advantage of not increasing the interpretation time. AI CADe takes 20% less time than traditional CADe, but the same amount of time as radiologists [57]. Although further studies are required to assess the exact costs of AI mammography, the overall reduction in false positives could make it cost-effective [57]. DL models are being incorporated into digital breast tomosynthesis, and contrast-enhanced digital mammography datasets for volumetric assessment of breasts in three dimensions to further increase detection accuracy and reduce workload by 70% [7,58,59]. Radiomics is an approach to extract relevant quantitative properties, also known as features, from clinical, histopathological, and radiological data. It has been applied to breast imaging to further improve accuracy rates [60]. A more detailed description of radiomics is described in Section 5.2.

### 3.2. Genetics and Hormonal Aspects in Breast Cancer Prediction

Artificial neural networks (ANNs) achieved remarkable accuracy, measured by AUC of 0.909, 0.886, and 0.883, when assessed for their ability to predict 5-, 10-, 15-year BC-related survival rates, respectively, based on factors such as age, tumor size, axillary nodal status, histological type, mitotic count, nuclear pleomorphism, and axillary nodal status [61]. Hybrid-DL models incorporate genetics, histopathology, and radiology data, which outperform traditional models such as Gail (which calculates BC risk in the next five years based on medical and reproductive history, not takes into account BRCA gene association) and Tyrer–Cuzick models (calculates the likelihood of carrying BRCA1 or BRCA2 mutations based on personal and familial historical data) [5,6].

## 4. Colonic Polyps and Colorectal Cancer

CRC is the third most common cancer in the United States, with the incidence of approximately 147,950 new cases in the year 2020. AI has shown great success in screening, diagnosis, and treatment of CRC. AI is bringing about a new era for CRC screening and detection with computer-assisted techniques for adenoma detection and characterization, computer-aided drug delivery techniques, and robotic surgery. Other benefits of AI include the incorporation of ANN to effectively screen with personal health data [62]. 

### 4.1. Colorectal Cancer Screening

By detecting adenomas and preventing progression to carcinoma, screening has significantly reduced the incidence of CRC over the past decade. This has resulted in recommendations for routine screening starting at age 45 [63]. The current screening methods for CRCs include invasive procedures (colonoscopy (gold standard) and flexible sigmoidoscopy), minimally invasive procedures (capsular endoscopy), and non-invasive procedures (CT colonography or virtual colonoscopy, stool for occult blood, fecal immunochemical test, and multitarget stool DNA). 

A few AI models have been tested to predict the risk of CRC and high-risk colonic polyps (CPs) from historical data and complete blood counts (CBCs). One such software, ColonFLag, predicts polyps and CRCs according to age, sex, CBC, and demographic information. Scores were compared to gold standard colonoscopy and converted to percentiles, then categories were made, such as CRC, high-risk polyps, and benign polyps [64]. Another retrospective study (MeScore, Calgary, Alberta, Canada) compared CBC results 3–6 months before colonoscopy with those from colonoscopy in two unrelated groups (Israeli and the UK). AUC for CRC diagnosis was 0.82 ± 0.01. Specificity for 50% detection was 87 ± 2% a year before diagnosis and 85 ± 2% for localized cancers [65]. Study results point to the possibility of an early and noninvasive preliminary screening that can be integrated into electronic medical records to flag high-risk patients who can then be aggressively screened to balance the risks and benefits of colonoscopy in young people. Another ANN model designed to screen a large population based only on personal health information from big data also achieved optimal results [62]. However, these models are not currently practiced and require further validation for generalizability.

### 4.2. Colonic Polyps Detection

Colonoscopy is the gold standard invasive testing for the detection of colonic adenoma and CRC. An adenoma is the most common precancerous lesion. Adenoma detection rate (ADR) measures a gastroenterologist’s ability to detect an adenoma. ADR is inversely related to the adenoma miss rate and the risk of post-colonoscopy CRC. ADR ranges from 7% to 53%, while AMRs vary from 6% to 27% based on healthcare facilities. Several factors have been postulated to explain these differences, including quality of preprocedural bowel preparation, time of withdrawal, operator experience and training, procedure sedation, cecal intubation rate, visualization of flexures (blind spots), and use of image enhanced endoscopy and presence of flat or diminutive (less than 5 mm) and small (<10 mm but >5 mm) polyps. Studies show that endoscopists with higher ADR during screening colonoscopy are more effective in preventing subsequent CRC risk for patients [66,67]. 

In recent years, CADe and computer-aided diagnosis (CADx) systems have been developed to automate polyp detection during colonoscopy and further characterize them. Because of its ability to detect diminutive polyps, real-time AI-aided colonoscopy has a greater ADR than colonoscopy (OR 1.53, 95% CI 1.32–1.77; *p* < 0.001), derived from a metanalysis data [4,68,69]. An AI system, GI Genius, uses green squares to highlight suspicious lesions during a colonoscopy by generating a sound for each marker and displaying it as a video of the endoscopy. Several meta-analyses demonstrate excellent detection rates for polyp detection using AI-assisted algorithms with AUC 0.90, sensitivity 95%, and specificity 88% [8]. 

### 4.3. Colon Polyps Classification

AI-based classification of CP into cancerous vs. non-cancerous lesions on CT colonography and capsular endoscopy is a fascinating discovery. CT colonography differentiation by texture analysis based on gradient and curvature of high-order images and random forest models significantly improved the accuracy of the classification of CPs [70,71]. AI-assisted CAD model revealed an inverse correlation of CP sphericity with adenoma detection sensitivity and a direct correlation with adenoma detection accuracy. This model can effectively detect flat colonic lesions and CRCs on CT colonography [72]. Capsule endoscopy is another noninvasive diagnostic tool for gastrointestinal tract inspection, but it is a time-consuming process to process a large amount of data. Stack sparse autoencoding with image manifold constraint, a DL-based AI, is utilized to correctly identify capsular polyps from capsular endoscopic images with a rate of 98% accuracy and time effectiveness [73]. An ANN model with logistic regression showed a predictive risk of distant metastasis in CRC patients based on several clinical factors, such as pathologic stage grouping, first treatment, sex, age at diagnosis, ethnicity, marital status, and high-risk behavior variables [74]. With DL models, tumors can be segmented and delineated more accurately, and faster region-based CNNs are trained to read MRI images, enabling faster and more accurate diagnosis of CRC metastasis [75,76].

### 4.4. Histopathological Aspects, Genetics, and Molecular Marker Detection

Histopathological characterization is the gold standard for the classification of polyps [77]. However, one of the biggest challenges is the significant intra- and interobserver variability. The use of DL and CNN models to automate image analysis can allow pathologists to classify CPs with an overall accuracy of 95% or more [10]. These DL models analyze whole slides and hematoxylin- and eosin-stained slides to identify four different stages, including normal mucosa, early preneoplastic lesions, adenomas, and cancer [9,10,78]. 

AI-based models were used to identify gene expressions, gene profiling, and non-coding micro-ribonucleotides (mi-RNAs) for diagnosis, prognosis, and targeted therapy planning [79,80,81]. The use of near-infrared (NIR) spectroscopy and counter propagation artificial neural networks (CP-ANNs) in the determination of mutant vs. wild B-rapidly accelerated fibrosarcoma (BRAF) gene mutations were shown to be highly accurate, specific, and sensitive [79]. Mutant BRAF is associated with a poor prognosis, and this AI model can assist in prognosticating and managing these patients aggressively. Backpropagation and learning vector quantization (LVQ) neural networks demonstrate a remarkable role in assessing the genetic profiling database from the cancer genome atlas (TCGA) in improving CRC diagnosis [81]. Several neural networks, including S-Kohonen, backpropagation, and SVM, were compared for predicting the risk of relapse after surgery. The S-Kohonen neural network was found to be the most accurate [82]. Non-coding mi-RNA plays an important role in tumorigenesis and progression of cancer by interfering with various cell signaling pathways, including, WNT/beta-catenin, phosphoinositide-3-kinase (PI3 K)/protein kinase B (Akt), epidermal growth factor receptor (EGFR), NOTCH1, mechanistic target of rapamycin (mTOR), and TP53. The identification of miRNAs through AI models aids in the diagnosis, prognosis, and targeted treatment of CRCs [80,83,84,85,86].

In the early detection of CRC, ML-based AI can help isolate circulating tumor cells in peripheral smear and analyze serum specific biomarkers, such as leucine-rich alpha-2-glycoprotein 1 (LRG1), EGFR, inter-alpha trypsin inhibitor heavy chain family member 4 (ITIH4), hemopexin (HPX), and superoxide dismutase 3 (SOD3) [87,88].

## 5. Central Nervous System Cancers

In the United States, primary brain tumors have an annual incidence of 14.8 per 100,000 people and have a male predominance. Despite significant advances in imaging modalities, surgical techniques, chemotherapy, radiotherapy, and radiosurgery, primary brain tumors such as glioblastoma multiforme (GBM) remains challenging to manage [89]. GBM is one of the primary intracranial neoplasms and accounts for nearly 60% of all primary brain tumors worldwide. Primary or metastatic CNS cancers are challenging to manage because of their rapid proliferation, prominent neovascularization, invasion to distant sites, and poor response to chemotherapy due to the blood–brain barrier. Clinical management includes initial observation, grading, accessing the depth of infiltration, segmentation and location of the tumor, histopathological evaluation, and identification of molecular markers. As a result, clinicians have to manually compile all the data for validation in order to formulate a treatment plan. In this regard, AI has proven to be useful in the diagnosis and management of CNS malignancies [26].

### 5.1. Central Nervous System Neoplasm Detection

AI has made significant advances in the diagnosis and classification of brain tumors in recent years. MRI is currently the gold standard tool for tumor detection and characterization [90]. Conventional MRI methods such as T_1_ and T_2_ weighted imaging and fluid-attenuated inversion recovery (FLAIR) sequences have the disadvantage of nonspecific contrast enhancement and a high likelihood of missing tumor foci infiltration. In order to enhance detection chances, perfusion MRI with dynamic susceptibility-weighted contrast material enhancement, dynamic contrast enhancement, and arterial spin labeling are also used to evaluate the neoangiogenic properties of brain tumors such as GBM. In addition to identifying tissue microstructure, diffusion-weighted imaging shows neoplastic infiltration in areas of the brain that appear normal on conventional magnetic resonance (MR) images. The use of MR spectroscopy can also be used to identify chemical metabolites such as choline, creatine, and N-acetyl aspartate, which are useful for glioma grading and identifying tumor infiltrated regions [91]. By automating these steps, AI has enhanced detection rates and efficiency of radiologists, which, in turn, has reduced the amount of time traditionally spent in diagnosing a disease. CNN-based DL can also detect millimeter-sized brain tumors and can distinguish GBMs from metastatic brain lesions [3,92]. MRI technologies provide structured anatomical information on tumors, but tumor differentiation is always based on histopathological evaluation, which is invasive, time-consuming, and expensive. It remains challenging to identify low-grade gliomas from high-grade gliomas on imaging, even with AI systems. Attention-based transformers are currently being investigated for the first time in glioma classification, and their use may offer a breakthrough [39,93].

### 5.2. Radiomics

A comprehensive analysis of clinical, histopathological, and radiological data combined with ML/DL image processing has paved the way for a new translational field in neuro-oncology called radiomics [60,94,95]. AI-based radiomics provides enhanced noninvasive tumor characterization by enabling histopathologic classification/grading within minutes even at surgery time, prognostication, monitoring, and treatment response evaluation [96,97]. AI algorithms are able to analyze these images at the pixel level, so they can provide information not visible to the human eye and allow for more accurate grading [3]. Radiomics involves a set of the complex multi-step processes with manual, automatic, and semi-automatic segmentations. Two main types of radiomics are described: feature-based and DL-based. Both provide more accurate and reliable results than human readers. The feature-based radiomics algorithms evaluate subsets of specific features from segmented regions and volumes of interest (VOI) into mathematical representations. This multistep process includes image pre-processing (noise reduction, spatial resampling, and intensity modification), precise tumor segmentation (manual vs. DL-based techniques), feature extraction (histogram-based, textural, and higher-order statistics features), feature selection (filter methods, wrapper approaches, and embedded techniques), and model generation and evaluation (neural networks, SVM, decision trees/ random forests, linear regression, and logistic regression models) [95,98]. DL radiomics use CNNs, in which the model learns in a cascading fashion without any prior description of features and requires a large amount of data in the learning process. The cascading technique processes data to obtain useful information, removes redundancies, and prevents overfitting [27,31,98]. 

### 5.3. Histopathological Aspects, Genetics, and Molecular Marker Detection

Traditional histopathological evaluation of cranial tumors identifies the microscopic features with areas of neovascularization, central necrosis, endothelial hyperplasia, and regions of infiltration. These are sometimes overlapping and could lead to false-positive results [99]. To overcome this complexity, digital slide scanners are now used to convert microscopic slides into image files interpreted by AI-based algorithms such as SVM and decision trees. SVMs have shown higher precision rates [98]. The AI-based algorithms analyze pathological specimens of gliomas and predict outcomes based on genetic and molecular markers, including isocitrate dehydrogenase (IDH) mutation status, 1 p/19 co-deletion status, O-6-methylguanine-DNA methyltransferase (MGMT) methylation status, epidermal growth factor receptor splice variant III (EGFRvIII), Ki-67 marker expression, prediction of p53 status in gliomas, prediction of mutations in BRAF, and catenin β-1 in craniopharyngiomas [96,98,100,101,102,103]. IDH mutation leads to the accumulation of an oncometabolite called D-2 hydroxyglutarate. This mutation is an important prognosticator in GBM. CNN-based AI has detected this biomarker from conventional MRI modalities [100]. O-6-MGMT promoter hypermethylation (encoding for DNA repair protein), which is exhibited in about 33%–57% diffuse gliomas, is a better prognostic factor owing to increased sensitivity to alkylating agents such as temozolomide [98,101,104]. AI types such as supervised machine learning combined with texture features have been found to detect this methylation status. Performing principal component analysis on the final layer of CNN indicated that features, such as nodular and heterogeneous enhancement and “masslike FLAIR edema”, predicted MGMT methylation status with up to 83% accuracy [105]. EGFRvIII mutation is found in about 40% of GBM. Tumors with this mutation have been found to exhibit deep peritumoral infiltration, which is consistent with a more aggressive phenotype. EGFR mutation is also associated with increased neovascularization and cell density [106]. 1 p/19 codeletion status has been shown to have a protective effect on the prognosis. This codeletion is observed in oligodendrogliomas [102]. CNN-based AI can be employed to detect this codeletion. Ki-67 marker expression indicates tumor cell proliferation. Traditionally, this marker is detected via immunohistochemical studies on the extracted tumor sample. This method is invasive and time-consuming. Identifying this marker is essential in making a differential diagnosis and treatment plan. AI-based radiomics has been developed to detect this marker from fluorodeoxyglucose positron emission tomography (FDG PET) and MRI images [107].

### 5.4. AI in Pre- and Intra-Operative Planning, Postoperative Follow-Up, and Metastasis

#### 5.4.1. Preoperative Assessment

Segmentation, volumetric assessment, and differentiating the tumor from healthy brain tissue and peripheral edema, quantitative measurements such as risk stratification, treatment response, and outcome prognosis are essential elements in the treatment planning of CNS tumors [108,109]. In traditional radiographic imaging, contrast-enhanced radiographic images are used to estimate tumor volume or burden; however, single-dimension imaging may not be as accurate in the volumetric assessment of nonuniform tumors, such as high-grade tumors including GBMs. Another challenge is differentiating tumor borders from surrounding edema [110]. AI algorithms such as the random forest, CNN, and SVM have been applied to the tumor segments to overcome these challenges, and they have been shown to provide precise and accurate localization of the tumor. A two-step protocol with CNN and transfer learning models led to precise and accurate localization of glioma [111]. 3D-U-Net CNN on 18 -fluoroethyl-tyrosine-PET, when used for automated segmentation of gliomas, showed 88% sensitivity, 78% positive prediction, 99% negative prediction, and 99% specificity [32,112]. 

#### 5.4.2. Intraoperative Modalities

High-grade tumors such as GBM have a rapid proliferation rate and invade the surrounding regions beyond the enhancing regions on the radiological images, and excision of these areas could be missed [26,113]. AI-based DL algorithms have been developed to facilitate the surgeons to remove maximum tumor regions and less of the normal healthy brain tissue simultaneously. Three-dimensional CNNs have shown promising results in aiding stereotactic radiation therapy planning. It is often difficult to differentiate among primary brain tumors, primary CNS lymphoma, and brain metastases in some situations. However, AI-based algorithms such as decision tree and multivariate logistic regression models have been developed to differentiate among these entities by using diffusion tensor imaging and dynamic susceptibility-weighted contrast-enhanced MRI [114,115,116].

#### 5.4.3. Postoperative Surveillance

MRI with gadolinium contrast is the standard for determining postoperative tumor growth and tumor response [117]. CNN-based AI algorithm techniques determine accurate tumor size compared to linear methods. The ability of CNN models to differentiate the true progression from pseudo-progression and ML algorithms to differentiate radiation necrosis from tumor recurrence is revolutionary [109,110,118]. Additionally, CNN and SVM create a superior model to predict the treatment response and survival outcomes from clinical, imaging, genetic, and molecular marker data [26].

## 6. Precision and Personalized Medicine

AI has moved towards an era of personalized treatment in oncology with remarkable aid in oncologic drug development, clinical decision support systems, chemotherapy, immunotherapy, and radiation therapy [43]. AI algorithms have been developed to assess several factors such as oncogenetic mutation profile and drug sensitivity prediction showing overall expected prognosis, efficacy, and adverse effects with a particular treatment option in a patient with particular cancer [43,119]. In a study, an ML algorithm was designed to predict the effects of chemotherapy drugs, including gemcitabine and taxols, in correlation to patients’ genetic signatures [120]. In another study, an AI-based screening system based on homologous recombination (HR) deficiency was developed to detect cancer cells with HR defects can further narrow patients who would benefit from poly ADP-ribose polymerase (PARP) inhibitors in BC patients [44]. A DL algorithm was used to identify anticancer drugs that inhibit PI3K alpha and tankyrase, promising targets for CRC treatment [121]. An ML-based drug specificity detection by examining protein–protein interactions of anticancer drug and S100A9, a calcium-binding protein, may represent a potential therapeutic target for CRC [122]. These avenues of discovery of new anticancer targeted therapy by ML models is a fascinating step towards much effective therapeutic options. ML models can also be trained to interpret screening data to predict responses to new drugs or combinational therapies [123]. An ability to synthesize and assess a large amount of chemical data also plays a role in cancer drug development by narrowing the prediction towards a specific formula; beyond the traditional experimental methods in which DL systems are currently being explored [124,125]. Learning clinical big data of cancer patients with AI can generate personalized treatment options based on DL assessed factors, including clinical, genetic, cancer-type, and stage of cancer of a patient [126]. Moreover, AI application in radiotherapy is quite distinct. AI can help radiologists plan radiation treatment regimens with automation software as effective as conventional treatment layouts in a robust, time-effective manner [127,128]. With the upcoming role of immunotherapy in managing various cancers, ML-based platforms are trained to predict the therapeutic response of immunotherapy effects in programmed cell death protein 1 (PD-1) sensitive advanced solid tumors [129,130]. AI can thus support and even surpass the capability of humans in anticancer drug development and aid in personalized treatment plans in a time-effective manner.

## 7. Generalizing Artificial Intelligence, Barriers, and Future Directions

A number of factors challenge the generalizability of AI systems, including possible bias, external validation of AI performance, the requirement for heterogeneous data and standardized techniques [46]. 

### 7.1. AI Performance Interpretation

In order for AI to perform in clinical practice, it must be both internally and externally validated. In internal validation, the accuracy of AI is compared to expected results when AI algorithms are tested by using previously used questions [131]. Internal validation performance tools rely on sensitivity, specificity, and AUC. The problem with interpreting AUC is that it does not consider the clinical context. For instance, different sensitivity and specificity can provide similar AUCs. In order to measure AI performance, studies should report AUC along with sensitivities and specificities at clinically relevant thresholds, this is referred to as “net benefit” [132]. As an example, high false-positive and false-negative rates continue to be a challenge in DL screening mammograms, for which balancing the net benefit would be important [42]. Thus, prior to concluding that an AI system can outperform a human reader, it is important to carefully interpret its diagnostic performance. Furthermore, the sensitivity, specificity, and accuracy of diagnostic tests are independent of real-life prevalence. As a result, robust clinical diagnostic, and predictive performance verification of AI for clinical applicability requires external validation. For external validation, a representative patient population and prospectively collected data would be necessary to train AI algorithms [131]. Moreover, internal validation poses the challenge of overestimating AI performance by familiarizing itself too much with training data, known as overfitting [131]. By separating unused training datasets, including newly recruited patients, and comparing results with those of independent investigators at different sites, it is possible to improve generalizability and minimize overfitting [131]. In a recent study, curated large mammogram screening datasets from the UK and the US revealed a promising path to generalizing AI performance [55]. 

### 7.2. Standardization of Techniques

An AI model that could be universally applicable must be taught a large amount of heterogeneous clinical data in order to become generalizable [3,54,107]. AI-based infrastructure and data storage systems are not available at all institutes, which is one of the biggest barriers [133]. There is also a lack of standardization of staining reagents, protocols, and section thicknesses of radiologic images, which can further hinder the generalizability of AI in clinical practice worldwide [1,54]. A number of automated CNN-based tools such as HistoQC, Deep Focus, and GAN-based image generators are being developed by societies such as the American College of Radiology Data Science Institute to standardize image sections [1,91]. In the field of radiomics, another challenge involves compliance with appropriate quality controls, ranging from image processing to feature extraction and from mechanics and feature extraction to algorithms for making predictions [134]. There are several emerging initiatives using DLs and CNNs to normalize or standardize images, including, “image biomarker standardization technique” [134,135]. ML algorithms are treated as a “black box” because of a lack of understanding of its inner working. This can pose a challenge when dealing with regulated healthcare data. This necessitates transparent AI algorithms and the interpretation of AI-based results to ensure no mistakes are made [26,136]. A few recently developed methods, such as saliency maps and principal component analysis, are helping interpret the workings of these algorithms [105,137].

### 7.3. Bias in Artificial Intelligence

Quality and quantity of data are key factors that determine the performance and objectivity of an ML system. AI can be biased in a number of ways—from assumptions made by engineers who develop AI to bias in the data used to train it. When training data are derived from a homogenous population, they may be poorly generalizable, which can potentially exacerbate racial/ethnic disparities, for example [138]. Thus, when training the AI, it is important to include diverse ethnic, age, and sex groups, as well as examples of benign and malignant tumors. Similarly, to integrate precision medicine and AI in real-world clinical settings, it is necessary to consider environmental factors, limitations of care in resource-poor locations, and co-morbidities [139]. There is also the possibility of bias introduced when radiologists’ opinion is regarded as the “gold standard” rather than the actual ground truth or the absolute outcome of the case, benign or malignant [46]. As an example, several AI models in screening mammography are compared with radiologists instead of the gold standard biopsy results, introducing bias [46]. In order to overcome this problem, including interval cancers in testing sets and relying on reports from experienced radiologists might be helpful.

### 7.4. Ethical and Legal Perspectives

Creating future models that address the ethical issues and challenges of incorporating AI into preexisting systems requires an awareness of these issues. Few societies, such as the Department of Health and Social Care, the US Food and Drug Administration, and other global partnerships, oversee and regulate the use of AI in medicine [46,140]. The National Health Service (NHS) Trusts in the United Kingdom regulate the use of patient care data in AI in an anonymized format for research purposes [46]. In order for AI in oncology to achieve global standardization, more international organizations must be formed that can oversee future AI studies within ethical and legal boundaries to protect patient privacy.

## 8. Integrative Training of Computer Science and Medical Professionals

In order for AI to be effectively integrated into healthcare in general, as well as oncology, formal training of medical professionals and researchers would be critical. Numerous societies and reviews have recommended formal training, but current medical education and health informatics standards do not include mandatory AI education, and competency standards have yet to be established [141,142]. There have been efforts in the radiology community to determine students’ opinions about AI applications in radiology in order to develop formal training tools. A few of these are frameworks for teaching, principles for regulating the use of AI tools, special training for evaluating AI technology, and integrating computer science, health informatics, and statistics curriculum during medical school [143,144,145]. Few institutes in the United States have proposed initiatives for AI in medical education, which were originally submitted by the American Medical Association. Among these initiatives are medical students working with data specialists, radiology residents working with technology base companies to develop computer-aided detection in mammography, offering a summer course by scientists or engineers to update new technologies, and involving medical students in engineering labs to create innovative ideas in health care [136]. Another framework would provide AI training for students in various fields, including medical students, health informatics students, and computer science students [142]. In order to improve patient care, medical students should become proficient in interpreting AI technologies, comparing efficiency in patient care and discussing ethical issues related to using AI tools [142]. Furthermore, medical professionals should understand the limitations and barriers of AI in clinical applications, as well as the distinction between correct and incorrect information [146,147]. In health informatics, students should be taught how to apply appropriate ML algorithms to analyze complicated medical data, integrate data analytics, and formulate questions to visualize large data sets. Students studying computer science should be trained in Python, R, and SQL programming in order to solve complex medical problems [142]. Education tools that integrate medical professionals, health informatics students, and computer science students can pave the way for further developments in the fields of medicine and oncology.

## 9. Conclusions

Computer systems are capable of learning tasks and predicting outcomes without being explicitly programmed through AI. DL, a subset of ML, utilizes neural networks and enables learning complex, non-linear functions from data. CNNs are well suited to process two- to three-dimensional inputs such as images, while RNNs can handle sequential inputs of variable length such as textual data. Recently developed attention-based DL systems are capable of selectively focusing on data, resulting in better accuracy in cancer detection rates. AI has shown promising results in oncology in several areas, including detection and classification, molecular characterization of tumors, cancer genetics, drug discovery, predicting treatment outcomes and survival rates, and moving the trend towards personalized medicine. In screening mammography, various DL models have demonstrated non-inferior cancer detection performance, with overall sensitivity rates of 88–96%. Radiologists with AI-assisted systems have achieved higher AUC rates and have reduced their workloads. Different real time CADe and CADx AI systems have demonstrated a higher ADR by automating polyp detection and detecting diminutive polyps during colonoscopy. The use of machines to improve cancer detection at an early stage on screening mammograms and colonoscopies has the potential to be tested for application across the globe for more efficient patient care. Several AI-based cancer detection methods have been developed for other cancer types, including lung, prostate, and cervical cancer. It is possible to pursue future objectives to implement AI worldwide in all cancer types. 

CNS tumors such as GBM continue to have a poor prognosis. AI-based radiomics allows for the identification of tumors without invasive methods, by allowing for the classification and grading of tumors within minutes. Radiomics is largely used in CNS tumors identification and grading. State-of-the-art attention-based transformers are currently being studied to improve glioma classification. Analyzing histopathological, genetic, or molecular markers can be made easier with AI. With the advancement of AI, oncology has moved to a more personalized era. AI has revolutionized drug development, clinical decision support systems, chemotherapy, immunotherapy, and radiotherapy. 

A better understanding of the ethical implications of the use of AI, including its performance interpretation, standardization of techniques, and the identification and correction of bias, is required for more reliable, accurate, and generalizable AI models. Global organizations must be formed to provide guidance and regulation of AI in oncology. Formal integrated training for medical, health informatics, and computer science students could drive further advances of AI in medicine and oncology.

## Figures and Tables

**Figure 1 cancers-14-01349-f001:**
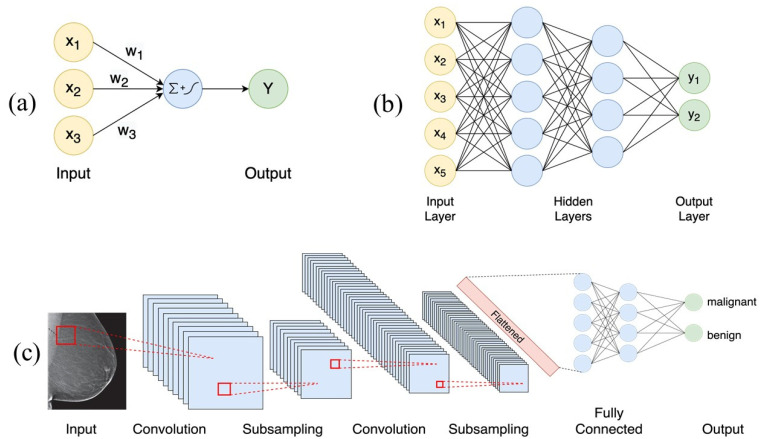
(**a**): Neuron, the fundamental computational unit of a neural network, computes the weighted sum of its inputs (X_1_, X_2_, X_3_) and applies a non-linear operation to give output (Y). (**b**): An example of a feedforward neural network with two hidden layers, with five and four neurons, respectively. (**c**): An example of a convolutional neural network (CNN) applied to the classification of a screening mammogram as probable malignant or benign.

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
