# Peer review of "Advancements in Oncology with Artificial Intelligence—A Review Article"

_cancers, 2022, doi:10.3390/cancers14051349_

Round 1

Reviewer 1 Report

The manuscript looks more complete and can be accepted in its current form except for some minor corrections with spellings.

Best regards

Reviewer 2 Report

Authors have replied to the second round of revisions.

Reviewer 3 Report

The authors made significant changes to the paper and have addressed my concerns. A few more additional remarks follow.

Abstract, first sentence: vague, long and convoluted, including 3 "and"s, not clear what the take away message is. Please make more concise and precise.

--> for example something like: "Machine learning (ML) and artificial intelligence (AI) systems can provide clinicians with therapeutic assistance."

Maybe you can state that it has potential to increase efficiency but also be more efficacy of therapies?

One further avenue that might be of interest, are the recent developments in 3D segmentation. CNNs have computational limitations in this regard and recently more efficient 3D methods have been proposed. O-Net the derived OSS-Net use function space representations for more efficient segmentation, for example.

Author Response

This manuscript is a resubmission of an earlier submission. The following is a list of the peer review reports and author responses from that submission.

Round 1

Reviewer 1 Report

The authors here, present a manuscript reviewing the application and advancements of Artificial Intelligence (AI) in Oncology, the current trends, and challenges, highlighting the need for standardization in data acquisition.

The Manuscript is well written, explaining AI, Machine learning before diving into their application in various Oncological disorders, including precision and personalized medicine.

It would also be interesting if the authors suggested possible solutions to overcome the barriers and challenges mentioned.

Thank you.

Author Response

The authors here, present a manuscript reviewing the application and advancements of Artificial Intelligence (AI) in Oncology, the current trends, and challenges, highlighting the need for standardization in data acquisition.

The Manuscript is well written, explaining AI, Machine learning before diving into their application in various Oncological disorders, including precision and personalized medicine.

It would also be interesting if the authors suggested possible solutions to overcome the barriers and challenges mentioned.

Thank you.

Response:

Thank you for reviewing our manuscript. We appreciate you bringing up an excellent point to highlight the possible solutions to overcome these barriers and challenges. We have updated these under section 7 in revised version with following information.

“7. Generalizing Artificial Intelligence, Barriers, and Future Directions:

A challenge to ethical and legal issues of generalizability of AI lies in the external validation of AI performance, the requirement for heterogeneous data and standard-ized techniques, and any potential bias in the systems [42].

7.1. AI Performance Interpretation: In order for AI to perform in clinical practice, it must be both internally and externally validated. In internal validation, the accuracy of AI is compared to expected results when AI algorithms are tested by using previously used questions [127]. Internal validation performance tools rely on sensitivity, speci-ficity, and AUC. The problem with interpreting AUC is that it doesn't consider the clinical context. For instance, different sensitivity and specificity can provide similar AUCs. In order to measure AI performance, studies should report AUC along with sensitivities and specificities at clinically relevant thresholds, which is referred to as "net benefit" [128]. As an example, high false-positive and false-negative rates continue to be a challenge in DL screening mammograms, for which balancing the net benefit would be important [38]. Thus, prior to concluding that an AI system can outperform a human reader, it is important to carefully interpret its diagnostic performances. Fur-thermore, sensitivity, specificity, and accuracy of diagnostic tests are independent of real-life prevalence. As a result, robust clinical diagnostic, and predictive performance verification of AI for clinical applicability requires external validation. For external validation, a representative patient population and prospectively collected data would be necessary to train AI algorithms [127]. Moreover, internal validation poses a chal-lenge of overestimating AI performance by familiarizing itself too much with training data, known as overfitting [127]. By separating unused training datasets, including newly recruited patients, and comparing results with those of independent investiga-tors at different sites, it is possible to improve generalizability and minimize overfitting [127]. In a recent study, curated large mammogram screening datasets from the UK and the US revealed a promising path to generalizing AI performance [51]. 

7.2. Standardization of Techniques: An AI model that could be universally appli-cable must be taught a large amount of heterogeneous clinical data in order to become generalizable [3, 50, 103]. AI-based infrastructure and data storage systems are not available at all institutes, which is one of the biggest barriers [129]. There is also a lack of standardization of staining reagents, protocols, and section thicknesses of radiologic images, which can further hinder the generalizability of AI in clinical practice world-wide [1, 50]. A number of automated CNN-based tools like HistoQC, Deep Focus, GAN-based image generators are being developed by societies such as the American College of Radiology Data Science Institute to standardize image sections [1, 87]. In the field of radiomics, another challenge involves compliance with appropriate quality controls, ranging from image processing to feature extraction and from mechanics and feature extraction to algorithms for making predictions [130]. There are several emerging initiatives using DLs and CNNs to normalize or standardize images, in-cluding, “image biomarker standardization technique” [130, 131]. ML algorithms are treated as a “black box”, because of a lack of understanding of its inner working. This can pose a challenge when dealing with regulated healthcare data. This necessitates transparent AI algorithms and the interpretation of AI based results to ensure no mis-takes are made [26, 132]. A few recently developed methods, such as saliency maps and principal component analysis, are helping interpret the workings of these algorithms [101, 133].

7.3. Bias in Artificial Intelligence: Quality and quantity of data are key factors that determine the performance and objectivity of an ML system. AI can be biased in a number of ways - from assumptions made by engineers who develop AI to bias in the data used to train it. When training data is derived from a homogenous population, they may be poorly generalizable, which can potentially exacerbate racial/ethnic dis-parities, for example [134]. Thus, when training the AI, it is essential to include diverse ethnic, age, and sex groups, as well as examples of benign and malignant tumors. Similarly, to integrate precision medicine and AI in real-world clinical settings, it is necessary to consider environmental factors, limitations of care in resource-poor loca-tions, and co-morbidities [135]. There is also the possibility of bias introduced when radiologist's opinion is regarded as the "gold standard" rather than the actual ground truth or the absolute outcome of the case, benign or malignant [42]. As an example, several AI models in screening mammography are compared with radiologists instead of the gold standard biopsy results, introducing bias [42]. In order to overcome this problem, including interval cancers in testing sets and relying on reports from experienced radiologists might be helpful.

7.4. Ethical and Legal Perspectives: Vigilance towards potential ethical questions and challenges of incorporating AI into preexisting systems is essential to create future models addressing these barriers. Few societies, such as the Department of Health and Social Care, the US Food and Drug Administration, and other global partnerships, oversee and regulate the use of AI in medicine [42, 136]. The National Health Service (NHS) Trusts in the United Kingdom regulate the use of patient care data in AI in an anonymized format for research purposes [42]. In order for AI in oncology to achieve global standardization, more international organizations must be formed that can oversee future AI studies within ethical and legal boundaries to protect patient privacy.”

Thank you for your time and consideration.

Reviewer 2 Report

The manuscript "Advancements in Oncology with Artificial Intelligence - A Review Article" by Vobugari et al. reviews artificial intelligence and its impact and potential to oncology. The manuscript describes artificial intelligence techniques and their application to oncology and includes many recent articles. The manuscript is well written, the structure requires modification for clarity. The methodology of the literature review is not clear, nor are the key conclusions. For example, while the importance of ethical, legal and economic aspects is highlighted in the abstract, these are not discussed in detail and only mentioned again in the conclusion. The paper claims AI education to be important for a wide audience, however, does not go into sufficient detail or depth on key concepts such as training of ML networks. The topic is interesting to a wide audience of Cancers readers; however, the manuscript requires major revisions and additions to be accessible and insightful to that audience. I am confident that the authors can make major revisions to the manuscript, to the extent that it  become suitable for publication in Cancers.

Key suggestions for improvement:

Add description of training methodology and procedures to answer the question of how DL works (which is posed in the section 2 title). Adding a description of training and gradient decent / back propagation would be beneficial to enable a broader audience access to the topic (which seems to be the core goal of the paper). This should include a detailed account of training bias, which is references frequently in the manuscript, yet never explained. For example, line 429f. speak of bias stemming from ‘biased minds’. This requires a deeper discussion of bias, including the potential challenges stemming from unbalanced training sets. The discussion of encountered bias (line 431) should be expanded for clarity.

I propose rewriting the conclusion, focussing on the key points discussed in the manuscript, and/or discuss the points presented in the conclusion in detail in the manuscript (see below for details).

It is not clear how the conclusions are made, as many of them are not discussed in the manuscript, little to no evidence is given in their support. The discussed knowledge gaps are unclear, the contributions of AI and the existing challenges are not clear in the conclusion; please clarify. Legal, ethical and economic aspects are mentioned in the abstract and conclusion, however, never really discussed in the body of the paper; please clarify and discuss in detail. The conclusion proposes the introduction of AI to the medical syllabus, however, nowhere is this discussed in the manuscript, and the manuscript itself does not really introduce the key AI, ML, or DL concepts to a sufficient depth or level of detail; please add.

CNNs should be explained in the general section and not relegated to the breast cancers section. CNNs appear throughout the review, however, the concept is introduced in a section dedicated to breast cancer.

Add some methodology on how the literature review was performed? How were the topics of the sections chosen, why were some cancers discussed, while others aren’t etc.?

‘Over the past four decades, the incidence and death rates of BC has decreased annually from 1989 to 2017, which can be attributed to the advancements in screening and therapies. (line 119)’ 1989 to 2017 is 28 years, calling it 4 decades is debatable, please reconsider. More importantly, how can improved screening reduce the incidence of a cancer? Please clarify and correct accordingly.

Many statements and claims made without evidence (eg. citations), examples include line 47.ff, 242ff. Please review all statements and claims and provide appropriate evidence.

Highlighting the recent advances in AI that have not yet made their way into oncology would contribute to the outlook of the paper. There are many more techniques that exist today and may become relevant to oncology in the future. The readership may benefit from being introduced to the most promising of these. For example, attention-based transformers have recently demonstrated state of the art instance segmentation performance and have been used for the first time to segment cells (presented at IEEE BIBM 2020). Their application may be beneficial in oncology.

Minor comments:

- line 27ff.: the first ‘are’ in the sentence seems out of place, consider rewriting the sentence completely.

- line 32: unclear: ‘more effective management.’ Of what?

- line 36: unclear: ‘potential trainers bias’. Please clarify what exactly is meant.

- line 47f.: please supply supporting evidence for claim (eg. citation), also for the other claims in the paragraph

- line 90: ReLU, abbreviation not written out

‘BC is the most common cancer worldwide, excluding skin cancers, and is a major cause of cancer-related mortality after lung cancer.  (line 117)’ If BC is not the most common cancer, then do not write that it is. Just say it is an important type of cancer, the second most prevalent. Could other factors be more important than prevalence, maybe impact on life expectancy?

- line 224: please explain CP abbreviation, this seems to be the first

- line 242: ‘Histopathological characterization is the gold standard for classification of polyps.’ please supply supporting evidence for claim (eg. citation), also for the other claims in the paragraph

Sdction 7: break into topical paragraphs.

Reviewer 3 Report

This work is a well-written review about AI and oncology.

If I understand well the review is focused on AI and tumor detection and prediction.  Sometime authors mention also AI performance in segmentation.

-The three types of cancer faced by the authors in this review which concerns AI methods in oncology, are quite different in the modality of techniques that are used to identify these tumors. This aspect gives to this review a wide scope and helps understanding the various potentialities of AI techniques; on the other hand the reader experiences the lack of a central idea. Could the authors explain the reason for the choice of describing these specific tumors (breast, colon, CNS)?

-As described in the Paragraph 7 there are many limitations in the AI approaches to oncology. Nevertheless, throught the description of all the different AI approaches for detection and prediction of breast, colon and CNS cancers, there are no descriptions of specific limitations concerning the literature reviewed. I would underline for the cited original researches the limitations and critical points of AI approaches in addition to their advantages.

-Concerning CNS cancers I would mention the topic of AI for tumor grade characterization  that is still a challenging objective, in particular the ability to differentiate low grade from high grade tumors.

Minor comments

Some references are missing: line 55 as regarding CNS.

Breast prognostication: some references missing:

  1. Overview of radiomics in breast cancer diagnosis and prognostication.

Tagliafico AS, Piana M, Schenone D, Lai R, Massone AM, Houssami N.

Breast. 2020 Feb;49:74-80. doi: 10.1016/j.breast.2019.10.018. Epub 2019 Nov 6.

  1. CAD and AI for breast cancer-recent development and challenges.

Chan HP, Samala RK, Hadjiiski LM. Br J Radiol. 2020 Apr;93(1108):20190580. doi: 10.1259/bjr.20190580. Epub 2019 Dec 16.

-Line 286: T1 and T2

Reviewer 4 Report

In this manuscript authors conduct a comprehensive review of the applications of AI in oncology. They focus, specifically, in how AI, DL and ML tools are implemented to improve or assist diagnostic, prognostic and therapeutic management in breast, colorectal and brain cancers.

Unarguably, the manuscript is well written, the bibliographic research is extensive and the issue is of high relevance for the general audience. However, this review lacks of a critical point of view, which is only shortly developed at the end of the manuscript. In addition is not justified anywhere why authors decided to write an article about the applications of AI in this three types of cancers. There are not common points nor links between the three presented disciplines and it seems a compilation of current evidence in these fields. Undoubtedly authors know that the current body of knowledge for each of these cancers would be more than enough to write a review article. Consequently, given that there are not common points to establish a comprehensive  discussion, there is no reason to present these three broad topics altogether.

I would recommend the authors considering to thoroughly review their article, expand the bibliographic review, deepen into the future challenges (ethic, universalization, technology), limitations and applications of AI, and separately write a manuscript for each of these three sub-themes.

Regarding CNS tumors there is an open special issue in Medicina, an MDPI publication, in which such an article could fit:

https://www.mdpi.com/journal/medicina/special_issues/Brain_Tumors_Central_Nervous_System_CNS_AI

Some minor points:

-Repeated:

CNNs are designed to capture spatial context and learn correlations between local features, due to which they yield superior performance on image tasks. CNNs are designed to capture spatial context and learn correlations between local features, due to which they yield superior performance on image tasks, such as, classification of breast lesions in a screening mammogram as probable malignant or benign

-GBM is a preferred abbreviation for glioblastoma.

-In line 332: Brain carcinoma is not an accurate definition for glial tumors. Molecular biology described after this term in the manuscript refers to current biomarkers for gliomas ,therefore I guess this is a mistake.

-Tumor burden might not be appropriate for brain tumors. AI algorithms are used to delineate or define tumor borders which in some cases, specifically in infiltrative gliomas (WHO II-IV), might be very challenging and may require various MRI sequences to distinguish, for instance, edema from tumor and radionecrosis from recurrence.

Round 2

Reviewer 2 Report

The authors made significant changes to the paper and have addressed my concerns.

That being said, I would encourage toning down some of the claims ('imperative', 'essential' in the conclusion for example).

Reviewer 3 Report

The authors have extensively reviewed their manuscript following my suggestions. 

At line 335 again: T1 and T2 have to be written using the subscript for 1 and 2.

Reviewer 4 Report

Authors have addressed my suggestions and the paper has been updated with such modifications.